# Non-Saccular Aneurysm Shape as a Poor Prognostic Factor in Younger Patients with Spontaneous Subarachnoid Hemorrhage

**DOI:** 10.3390/jcm14124289

**Published:** 2025-06-16

**Authors:** Fumihiro Hamada, Hitoshi Fukuda, Yuma Hosokawa, Shota Nishimoto, Yuichiro Kondo, Masaki Yokodani, Koji Bando, Yu Hoashi, Kenji Okada, Akihito Moriki, Takahiro Niimura, Nobuhisa Matsushita, Yo Nishimoto, Maki Fukuda, Motonobu Nonaka, Yu Kawanishi, Yusuke Ueba, Naoki Fukui, Tetsuya Ueba

**Affiliations:** 1Department of Neurosurgery, Kochi Medical School, Kochi University, Nankoku 783-8505, Japan; hamada-fumihiro@kochi-u.ac.jp (F.H.); jm-yhosokaway@kochi-u.ac.jp (Y.H.); mtb.nonaka@gmail.com (M.N.); ykawanishi@kochi-u.ac.jp (Y.K.); yueba@kochi-u.ac.jp (Y.U.); naofukui@kochi-u.ac.jp (N.F.); tueba@kochi-u.ac.jp (T.U.); 2Department of Neurosurgery, Chikamori Hospital, Kochi 780-0052, Japan; ns7566u@yahoo.co.jp (S.N.); yoh.nishimoto@gmail.com (Y.N.); 3Department of Neurosurgery, Kochi Health Science Center, Kochi 781-0111, Japan; yuichirokondo0820@gmail.com; 4Department of Neurosurgery, Hata Kenmin Hospital, Sukumo 788-0785, Japan; yokodani4543@gmail.com (M.Y.); xythph@par.odn.ne.jp (M.F.); 5Department of Neurosurgery, Kochi Red Cross Hospital, Kochi 780-0026, Japan; kjbando0418@gmail.com (K.B.); nobuhisama@gmail.com (N.M.); 6Department of Neurosurgery, Izumino Hospital, Kochi 781-0011, Japan; hoashiyu@gmail.com; 7Department of Neurosurgery, Aki General Hospital, Aki 784-0027, Japan; kenjiokada1030@gmail.com; 8Department of Neurosurgery, Mominoki Hospital, Kochi 780-0952, Japan; moriki1013@gmail.com; 9Clinical Research Center for Developmental Therapeutics, Tokushima University Hospital, Tokushima 770-0042, Japan; niimura@tokushima-u.ac.jp

**Keywords:** non-saccular aneurysm, subarachnoid hemorrhage, ischemic complications, functional outcomes, parent artery occlusion

## Abstract

**Background/Objectives:** Non-saccular aneurysms are a rare subtype of intracranial aneurysms with complex morphologies. Although treatment strategies for ruptured non-saccular and saccular aneurysms differ significantly, large-scale comparisons of the outcomes between the two types remain limited. We aimed to compare the clinical characteristics, procedure-related complications, and functional outcomes between patients with subarachnoid hemorrhage (SAH) caused by non-saccular or saccular aneurysms. **Methods**: We retrospectively analyzed 1176 consecutive patients with aneurysmal SAH from a population-based stroke registry in Kochi Prefecture, Japan. Aneurysms were classified as saccular or non-saccular based on the morphology, and clinical variables, radiological features, and treatment modalities were compared. Additionally, 840 patients who underwent intervention for their aneurysms within 3 days of onset were further investigated to evaluate the impact of the non-saccular aneurysm shape on poor functional outcomes, defined as a modified Rankin Scale score ≥ 3 at discharge. **Results**: Non-saccular aneurysms were more common in younger patients and located in the posterior circulation. Procedure-related ischemic complications were more likely to occur in non-saccular aneurysms than in saccular aneurysms (odds ratio [OR]: 2.57, 95% confidence interval [CI]: 1.56–4.97, *p* < 0.001). In a multivariable logistic regression analysis, a non-saccular morphology was an independent risk factor of poor outcomes (OR: 2.92, 95% CI: 1.34–6.32, *p* = 0.007) after adjustment for potential confounders. Interaction and subgroup analyses revealed that the negative effects of non-saccular aneurysms on functional outcomes were more prominent in younger patients aged ≤ 60 years. **Conclusions**: Non-saccular aneurysms are independently associated with ischemic complications and poor outcomes after SAH, particularly in younger patients.

## 1. Introduction

Intracranial aneurysms can be classified based on their pathophysiology, morphology, and etiology. Among these, morphological classification is the most common in clinical practice, typically dividing aneurysms into two main types: saccular and non-saccular aneurysms [1,2,3]. Although subarachnoid hemorrhage (SAH) caused by non-saccular aneurysms is relatively uncommon [4], these lesions are often more difficult to treat than saccular aneurysms because of their complex morphology [5]. Thus, a non-saccular aneurysm’s shape could be a determinant of poor prognosis of aneurysmal SAH, in addition to traditional risk factors, including older age and worse initial neurological status, through an increased incidence of procedure-related complications. However, the association between a non-saccular shape and poor SAH outcomes has been unclear to date because of its rarity, heterogeneity, and patients’ background distinct from those of patients with saccular aneurysms [1,2,3,4,5].

In this study, we investigate patient and aneurysm baseline characteristics, treatment modalities, procedure-related complications, and functional outcomes of SAH caused by intracranial non-saccular or saccular aneurysms. Specific features and treatment difficulties of ruptured non-saccular aneurysms are also discussed.

## 2. Materials and Methods

### 2.1. Ethical Considerations

This retrospective, multicenter observational study adhered to the STROBE (Strengthening the Reporting of Observational Studies in Epidemiology) guidelines. The study protocol was approved by the Research Ethics Committee of Kochi Medical School Hospital (approval number: #111099). The requirement for informed consent was waived due to the use of anonymized data lacking any personally identifiable information. The institutional review boards of all the participating hospitals also approved the protocol.

### 2.2. Patient Population and Study Design

Patient data were obtained from the Kochi Acute Stroke Survey of Onset (KATSUO) registry, an ongoing prefecture-wide stroke database established in 2011 and maintained by the Kochi Prefectural Office. The registry includes data from 29 hospitals within Kochi Prefecture. Owing to the closed nature of the regional emergency transport system, nearly all acute stroke cases in the prefecture are included, with inter-prefectural patient transfer occurring in only approximately 1% of cases [6].

This study focused on patients diagnosed with SAH and registered in the KATSUO registry between May 2011 and December 2022. The dataset, which was anonymized and provided by Kochi Medical School Hospital, initially included 1471 SAH cases from eight primary and comprehensive stroke centers that provided therapeutic intervention and neurocritical care for patients with SAH. Of them, 283 patients were excluded due to missing essential clinical information, including the initial neurological status represented by the World Federation of Neurosurgical Societies (WFNS) grade, discharge functional outcomes assessed using the modified Rankin Scale (mRS), subarachnoid clot volume based on the Fisher group, and bleeding source. An additional 12 patients were excluded for the following reasons: duplicate registration from an inter-hospital transfer (n = 5), alternative diagnoses such as arteriovenous malformations or infectious aneurysms (n = 3), intraoperative confirmation of unruptured aneurysms (n = 3), and prior treatment abroad (n = 1).

Of the 1176 patients included in the analysis of baseline characteristics, 336 were excluded from outcome-related analyses due to the absence of treatment, delayed intervention (initiated on or after day 4), baseline mRS ≥ 2, or recurrent SAH. The final cohort comprised 840 patients. Figure 1 illustrates the patient selection process. The baseline characteristics and clinical information of the patients with SAH due to ruptured non-saccular aneurysms were compared with those of patients with ruptured saccular aneurysms. The association of a non-saccular shape with SAH-related complications (symptomatic vasospasm, procedure-related complications, and chronic hydrocephalus) and functional outcomes was analyzed using univariate, multivariable, and subgroup analyses. The interaction effects of a non-saccular shape with traditional risk factors, such as age and WFNS grade, were also investigated.

### 2.3. Clinical Parameters and Data Collection

Patients were categorized based on aneurysmal morphology into saccular (n = 1077) and non-saccular (n = 99) groups based on the aneurysmal morphology. Non-saccular aneurysms were defined according to previously described definition of non-saccular aneurysm subtypes: dissecting aneurysms, fusiform aneurysms, and blood blister aneurysms [3,7,8,9,10]. Infectious aneurysms were excluded because a concomitant infection may affect functional outcomes. The registry provided detailed clinical and demographic variables, including age, sex, onset and admission dates, medical history (e.g., hypertension and smoking), WFNS grade, Fisher group, aneurysmal location (internal carotid artery [ICA], anterior cerebral artery [ACA], middle cerebral artery [MCA], or posterior circulation [PCQ]), treatment modality (endovascular therapy, direct surgery, or conservative management), symptomatic vasospasm, procedure-related complications, chronic hydrocephalus, and mRS score at discharge. The indication of therapeutic intervention, choice of treatment modality, and postoperative care were determined at the discretion of the attending physicians at each center.

Poor functional outcomes were defined as a discharge mRS ≥ 3 in patients with a pre-SAH mRS of 0 or 1. Procedure-related complications were defined as neurological worsening attributable to ischemia, hemorrhage, or clot progression. Symptomatic vasospasm was defined as a persistent neurological decline (lasting ≥ 8 h) associated with a reduction in the Glasgow Coma Scale of ≥2 points or worsening of the NIH Stroke Scale motor score of ≥2 points, with significant narrowing of the major vessels on digital subtraction angiography, computed tomography (CT) angiography, or magnetic resonance angiography, after excluding other potential causes by clinical and radiographic evaluation [11]. Chronic hydrocephalus was diagnosed as ventricular enlargement that required cerebrospinal fluid shunt placement.

### 2.4. Statistical Analysis

Continuous variables were summarized as means ± standard deviation and assessed using the Student’s *t*-test. Categorical variables were analyzed using the chi-square test. Univariate logistic regression analyses were conducted to identify the predictors of SAH-related complications. Univariate and multivariable logistic regression models were constructed and analyzed to investigate the association between a non-saccular shape and poor functional outcomes. Interactions between a non-saccular shape and other traditional risk factors were evaluated by including the interaction term in the final logistic regression model. Subgroup analyses were also performed by dividing the patient population by age and WFNS grade to determine the patient subgroups susceptible to the negative effects of a non-saccular shape on functional outcomes.

Missing values for comorbidities were observed in 52 patients with hypertension and 81 currently smoking and were complemented with multiple imputations to preserve statistical power and reduce bias. Odds ratios (ORs) and 95% confidence intervals (CIs) were determined in the logistic regression analyses. SPSS version 29 (IBM Corp., Armonk, NY, USA) was used for all the statistical analyses. Probability (*p*) values < 0.05 were considered statistically significant, and all *p*-values were two-sided. Furthermore, *p* < 0.10 was considered meaningful when interpreting the interaction term with respect to previous studies [12].

## 3. Results

Overall, 99 non-saccular and 1077 saccular aneurysm cases were analyzed. Table 1 compares the characteristics of non-saccular and saccular aneurysms. The mean age was significantly lower in the non-saccular group (62.3 ± 32.5 years) than in the saccular group (67.2 ± 30.2 years; *p* = 0.016). The proportion of female patients was significantly lower in the non-saccular group (56.6%) than in the saccular group (73.7%, *p* < 0.001). The prevalence of smoking was significantly different between the two groups (46.7% in non-saccular vs. 34.8% in saccular, *p* = 0.022). The aneurysmal location differed significantly between the groups (*p* < 0.001); non-saccular aneurysms were more frequently located in the posterior circulation (74/99, 74.7%) than saccular aneurysms (118/1077, 11.0%). Patients with a more severe initial neurological status (WFNS grades IV–V) were more likely to be observed in the non-saccular group (47.4%) than in the saccular group (41.8%). Endovascular treatment was significantly more common in the non-saccular group (56/99 cases, 56.6%) than in the saccular group (473/1077 cases, 43.9%), whereas direct surgery was more frequently performed for saccular aneurysms (48.5%) than for non-saccular aneurysms (17.2%). Adjunctive techniques, including parent artery occlusion (34.0% vs. 5.2%; *p* < 0.001), bypass surgery (15.1% vs. 0.7%; *p* < 0.001), and stent assistance (20.8% vs. 1.1%; *p* < 0.001), were more likely to be used in the non-saccular group than in the saccular group. Conservative treatment was more common in the non-saccular group (26.3% vs. 7.6% in the saccular group). Among cases involving conservative treatment, a better neurological status at admission (WFNS grades I–II) was observed more commonly in the non-saccular group (8/26, 30.8%) than in the saccular group (5/83, 6.0%) (*p* = 0.002), suggesting that conservative treatment was chosen for reasons other than poor neurological status at admission in the non-saccular group. A total of 99 non-saccular aneurysms were morphologically categorized into 62 dissecting, 19 fusiform, and 18 blood blister aneurysms. While dissecting and fusiform aneurysms were predominantly located in the posterior circulation, 14 of the 18 blood blister aneurysms were located in the ICA (Table 2).

Of the 1176 patients, 840 (53 non-saccular and 787 saccular groups) with a premorbid mRS of 0–1 and who experienced their first-ever SAH and underwent therapeutic intervention within 3 days after onset were analyzed for procedure-related complications, symptomatic vasospasm, chronic hydrocephalus, and functional outcomes. Of the 35 dissecting, 9 fusiform, and 9 blood blister aneurysms, 30, 3, and 5 were treated with endovascular therapy, and 14, 4, and 3 ischemic complications were observed, respectively, suggesting that dissecting aneurysms were more likely to be treated with endovascular therapy. The results of the univariate logistic regression analyses for the association of non-saccular aneurysms with procedure-related complications, symptomatic vasospasm, and chronic hydrocephalus are shown in Table 3. The incidence of ischemic complications was significantly higher in the non-saccular group than in the saccular group (39.6% vs. 19.1%, respectively; OR: 2.57, 95% CI: 1.56–4.97, *p* < 0.001). Ischemic complications occurred in 21 patients in the non-saccular group, of which 4 were ICA aneurysms and 17 were PCQ aneurysms. In contrast, hemorrhagic complications, symptomatic vasospasm, and shunt-dependent hydrocephalus were not significantly different between the two groups.

Univariate and multivariable logistic regression analyses were performed to identify independent predictors of poor functional outcomes (mRS 3–6) at discharge (Table 4). In univariate analyses, age, hypertension, smoking, WFNS grades IV-V, and Fisher CT group 3 were significantly associated with poor functional outcomes. A non-saccular aneurysm shape was not revealed as a significant risk factor of poor functional outcomes (OR: 1.23, 95% CI: 0.70–2.16; *p* = 0.47). However, in the multivariable analysis, a non-saccular shape remained as a significant predictor of poor functional outcomes after adjustment for potential confounders (OR: 2.90, 95% CI: 1.34–6.31; *p* = 0.007). Because of the differences in other traditional risk factors, including age and WFNS grade, between the non-saccular and saccular groups, interaction analyses for poor outcomes were performed. An interaction between a non-saccular shape and patient age was identified, and younger patients were more likely to be affected by the negative effect of a non-saccular shape than older patients (OR for interaction: 1.04, *p* = 0.099 [OR < 0.10 was predefined as meaningful in interaction analyses). No interaction effect was observed between a non-saccular shape and WFNS grades I–III (OR for interaction: 2.67; *p* = 0.14). Motivated by the result of the interaction analysis, a univariate logistic regression analysis of the association between a non-saccular shape and poor functional outcomes in the subgroup of the patients aged ≤ 60 years (341 patients) was performed, revealing that a non-saccular shape was a significant risk factor of poor functional outcomes in these patients (OR: 2.55, 95% CI: 1.24–5.22; *p* = 0.011).

## 4. Discussion

In this study, we identified a non-saccular aneurysm shape as an independent risk factor for poor functional outcomes in patients with SAH due to ruptured intracranial aneurysms. Procedure-related ischemic complications occurred more frequently in non-saccular aneurysms than in saccular aneurysms. Patients ≤ 60 years of age were more susceptible to the negative impact of a non-saccular shape on poor outcomes than older patients.

Most spontaneous (non-traumatic) SAHs are caused by the rupture of intracranial aneurysms. Although ruptured aneurysms typically have a saccular shape with a defined neck and dome, SAH is also caused by the rupture of non-saccular aneurysms without a clearly defined neck [2,13]. The characteristics of non-saccular aneurysms differ from those of saccular aneurysms in several respects. First, most saccular aneurysms are histologically true aneurysms, in which the disruption of internal elastic lamina and media is localized to a small part of the axial circumference of the vessel. In contrast, the formation of non-saccular aneurysms commences with a broader disruption of the internal elastic lamina, followed by various histological reactions, including blood flow into the pseudolumen, neointimal formation, thrombus formation, and partial disruption of the adventitia [14]. Second, the epidemiology is different; patients with non-saccular aneurysms are younger and include more males, and such aneurysms are more likely to be located in the posterior circulation [3,5,15,16,17,18]. Third, the natural history is different when left untreated. Unruptured non-saccular aneurysms rarely rupture and have a benign natural history, whereas unruptured saccular aneurysms rupture or cause adverse events with a certain probability [1]. The rebleeding rate of ruptured non-saccular aneurysms drastically decreases over several days and reaches almost zero after several months due to the healing process by neointimal formation, while ruptured saccular aneurysms retain a relatively high annual rebleeding rate for a lifetime [19,20]. Non-saccular aneurysms can be categorized into several subtypes according to their morphology and histology, including dissecting, blood-blister-like, fusiform, pseudo-, atherosclerotic, and dolichoectatic aneurysms, and have been investigated separately for their characteristics, treatment, and outcomes [3,7,8,9,10]. However, because these subtypes share a broader disruption of internal elastic lamina, epidemiology, and natural history, which are distinct from those of saccular aneurysms, and because SAH caused by non-saccular aneurysms is relatively uncommon, combining these subtypes into “non-saccular aneurysms” and comparing them with saccular aneurysms in terms of procedure-related complications and functional outcomes is meaningful [4].

In this study, procedure-related ischemic complications occurred at a significantly higher rate in non-saccular aneurysms than in saccular aneurysms. The ideal treatment to prevent a ruptured aneurysm from rebleeding is complete elimination of the bleeding source from the normal circulation, with preservation of the parent and perforating arteries. This can be achieved for saccular aneurysm, in which the defined neck is obliterated with neck clipping or intrasaccular coiling. However, such treatment is rarely available for the non-saccular aneurysms without a defined neck, and parent artery occlusion is often required to completely obliterate the bleeding point [21,22]. Parent artery occlusion was performed in 34.0% of the non-saccular aneurysms in our study and was associated with an increasing incidence of ischemic complications for several reasons. When the parent artery, particularly the ICA, is occluded for therapeutic purposes, the corresponding cerebral hemisphere becomes hemodynamically compromised, resulting in cerebral cortical infarction. Physiological collateral blood flow through the anterior and posterior communicating arteries and surgical construction of the vascular bypass may help prevent cortical infarction. However, it is unclear whether such collateral or bypass blood flow is large enough to counteract the delayed cerebral ischemia caused by vasospasm occurring several days after SAH onset [23]. In contrast, hemodynamic compromise is less likely to occur by parent artery occlusion of a non-saccular aneurysm of the vertebral artery (VA), because blood flow in the distal posterior circulation is compensated by the contralateral VA unless it is hypoplastic or occluded [22]. Nonetheless, parent artery occlusion of the VA, which is usually performed endovascularly, is predisposed to medullary infarction due to occlusion of the perforating arteries. Perforating arteries with an orifice at the aneurysmal wall or at the normal vessel wall across from the aneurysmal wall are involved in the coil mass. In addition, perforating arteries apart from the coil mass can be occluded by thromboembolism during coil embolization or stagnant flow in the parent artery after embolization [24]. Preservation of the patency of the parent artery by stent-assisted coiling or flow diverter placement may be promising for preventing ischemic complications associated with parent artery occlusion [25,26]. However, the use of a stent or flow diverter is prone to ischemic complications due to blood hypercoagulability as a systemic response and insufficient antiplatelet therapy in acute stages of SAH [26,27,28]. The relatively higher rate of incomplete or delayed obliteration of the stent or flow diverter treatment may be another concern in preventing rebleeding in the ultra-early stages of ruptured non-saccular aneurysms [26,28].

In this study, a non-saccular shape of the aneurysm was an independent risk factor for poor functional outcomes of aneurysmal SAH, and younger patients were more susceptible to the negative effects. Although an increasing age and poor initial neurological status have been revealed as major determinants of poor outcomes, the contribution of aneurysmal morphology remains debatable [29]. However, because the treatment strategy for a specific morphology of the aneurysm is modifiable, unlike age or initial neurological status, and because a non-saccular shape negatively affects younger patients, who would otherwise have favorable outcomes, avoidance of procedure-related complications by optimizing the treatment strategy is valuable. The optimal treatment strategy for ruptured non-saccular aneurysms may be modified according to their specific features. For example, the rebleeding rate of ruptured non-saccular aneurysms, including dissecting aneurysms, is extremely high within 24 h of onset but drastically decreases after several days and reaches almost zero in two months [19,20]. Thus, immediate and complete obliteration of ruptured non-saccular aneurysms in patients who arrive at the hospital on the day of SAH onset is recommended, especially for young patients who can tolerate acute invasive intervention and tend to recover well from ischemic complications [20]. However, when intervention for the aneurysm is delayed for some reason, including delayed visit of the patient, misdiagnosis, and challenging anatomy that does not allow for simple and immediate treatment, preservation of the parent artery by stent-assisted coil embolization or flow diverter placement may be beneficial. In the chronic stage, the relatively lower rebleeding rate of non-saccular aneurysms may allow for sufficiently effective delayed obliteration by stent or flow diverter. Delayed cerebral ischemia due to vasospasm and SAH-related hypercoagulability, which can cause thromboembolic complications associated with endovascular devices, subsides over time, and sufficient antiplatelet therapies are available [30]. Conservative treatment, by not treating but monitoring the aneurysm, may be an alternative approach, given the high probability of complete healing of ruptured non-saccular aneurysms at the chronic stage, particularly when treatment for the prevention of rebleeding is extremely challenging (e.g., basilar artery dissection).

This study had several limitations that warrant consideration. First the study’s observational, retrospective design of the study precludes causal inference; therefore, the findings should be interpreted with caution. Second, because the final analyses of functional outcomes only included patients who underwent aneurysmal treatment within three days after the onset, those who underwent delayed or conservative treatment were not investigated. Generally, the benign clinical course of patients with ruptured non-saccular aneurysms, who survived for several days without acute rebleeding, may impact the overall outcomes of the non-saccular aneurysm group. In addition, data regarding long-term functional outcomes and rebleeding events were unavailable, which hampered the evaluation of long-term functional recovery from ischemic complications of the patients and healing process of ruptured non-saccular aneurysms. However, to our knowledge, our study is the first to provide in-depth comparisons of procedure-related complications and functional outcomes of patients with ruptured non-saccular aneurysms using multivariable, interaction, and subgroup analyses with sufficient statistical power provided by a large, prefecture-wide registry in Japan. Further investigations, using longer-term follow-up data for example, are warranted to identify individualized optimal treatment strategies in this subgroup of patients with ruptured non-saccular aneurysms.

## 5. Conclusions

In conclusion, non-saccular aneurysms represent an independent risk factor for poor outcomes after SAH, presumably because of the increased incidence of ischemic complications. This risk is particularly pronounced in younger patients who would otherwise be expected to have favorable prognoses. These findings highlight the critical need for heightened clinical attention and potentially tailored management strategies for younger individuals with non-saccular aneurysms to mitigate the risk of adverse outcomes.

## Figures and Tables

**Figure 1 jcm-14-04289-f001:**
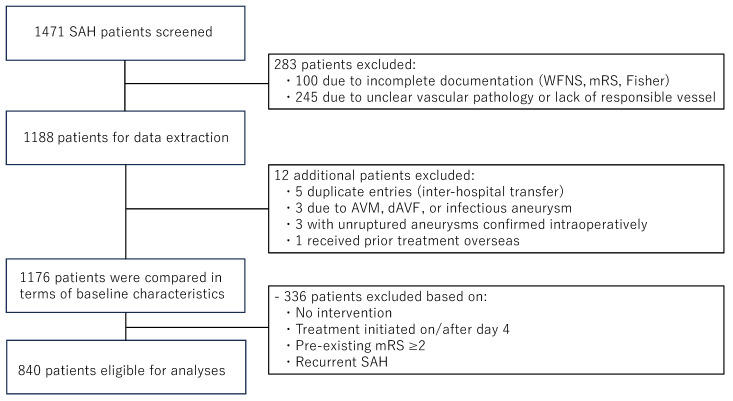
Patient selection flowchart.

**Table 1 jcm-14-04289-t001:** Comparison of characteristics between non-saccular and saccular aneurysms.

Variable	Non-Saccular	Saccular	Missing Value	*p*-Value
No. of patients	99	1077		
Mean Age (years)	62.3 ± 32.5	67.2 ± 30.2		**0.016**
Female sex	56 (56.6)	794 (73.7)		**<0.001**
Hypertension	62 (68.9)	624 (61.1)	52(4.4)	0.542
Smoking	43 (46.7)	349 (34.8)	81(7.5)	**0.022**
Recurrent SAH	11 (9.1)	76 (7.1)		0.454
Premorbid mRS ≥ 2	11 (9.1)	119 (11.0)		0.985
WFNS grade				**0.048**
I	21 (21.2)	344 (31.9)		
II	27 (27.3)	247 (22.9)		
III	4 (4.0)	36 (3.3)		
IV	13 (13.1)	194 (18.0)		
V	34 (34.3)	256 (23.8)		
Fisher group 3	83 (83.8)	877 (81.4)		0.554
Aneurysmal location				**<0.001**
ACA	0 (0)	381 (35.4)		
ICA	21 (21.2)	342 (31.8)		
MCA	4 (4.0)	236 (21.9)		
PCQ	74 (74.7)	118 (11.0)		
Intervention				**<0.001**
Direct surgery	17 (17.2)	522 (48.5)		
Endovascular therapy	56 (56.6)	473 (43.9)		
Conservative	26 (26.3)	82 (7.6)		

Values are expressed as mean ± 2SD or the number of patients (%). *p*-values were calculated using the *t*-test or chi-square test, as appropriate, and statistically significant valuables are indicated by boldface. ICA = internal carotid artery, ACA = anterior cerebral artery, MCA = middle cerebral artery, PCQ = posterior circulation, WFNS = World Federation of Neurosurgical Societies, mRS = modified Rankin Scale, SAH = subarachnoid hemorrhage.

**Table 2 jcm-14-04289-t002:** Morphological breakdown of the non-saccular aneurysms and their location.

	ACA	ICA	MCA	PCQ	Total
Dissecting	0	3	1	58	62
Fusiform	0	4	3	12	19
Blood blister	0	14	0	4	18
Total	0	21	4	74	99

ICA = internal carotid artery, ACA = anterior cerebral artery, MCA = middle cerebral artery, PCQ = posterior circulation.

**Table 3 jcm-14-04289-t003:** Univariate logistic regression analyses for procedure-related and SAH-related complications.

Variable	Non-Saccular	Saccular	OR [95% CI]	*p* Value
No. of patients	53 (6.3)	787 (93.7)		
Ischemic complications	21 (39.6)	150 (19.1)	2.57 [1.56–4.97]	**<0.001**
Hemorrhagic complications	3 (5.7)	28 (3.6)	1.63 [0.48–5.53]	0.44
Symptomatic vasospasm	2 (3.8)	101 (12.8)	0.27 [0.06–1.11]	0.07
Chronic hydrocephalus	14 (26.4)	166 (21.1)	1.34 [0.71–2.53]	0.36

Values are expressed as the number of patients (%). Statistically significant valuables are indicated by boldface. SAH = subarachnoid hemorrhage.

**Table 4 jcm-14-04289-t004:** Results of univariate and multivariable logistic regression analyses of risk factors for poor functional outcomes.

	Functional Outcomes		Univariate Analysis		Multivariable Analysis	
	Favorable	Poor	OR [95% CI]	*p*-Value	OR [95% CI]	*p*-Value
No. of patients	405	435				
Age (years)	58.6 ± 27.1	68.2 ± 27.7	1.05 (1.04–1.063)	**<** **0.001**	1.06 (1.05–1.08)	**<0.001**
Female sex	274 (67.7)	316 (72.6)	1.27 (0.94–1.71)	0.11	0.87 (0.57–1.32)	0.50
Hypertension	216 (53.3)	277 (63.7)	1.53 (1.16–2.03)	**0.003**	1.23 (0.87–1.75)	0.24
Smoking	190 (46.9)	149 (34.3)	0.56 (0.42–0.74)	**<** **0.001**	0.79 (0.52–1.18)	0.24
Non-saccular	23 (5.7)	30 (6.9)	1.23 (0.70–2.16)	0.47	2.90 (1.34–6.31)	**0.007**
WFNS 4-5	66 (16.3)	253 (58.2)	7.14 (5.16–9.89)	**<** **0.001**	9.66 (6.65–14.0)	**<0.001**
Fisher3	333 (82.2)	382 (87.8)	1.56 (1.06–2.29)	**0.02**	1.38 (0.87–2.20)	0.18
Location						
ICA	125 (30.9)	118 (27.1)	reference	0.64	reference	
ACA	141 (34.8)	157 (36.1)	1.18 (0.84–1.66)	0.34	1.50 (0.98–2.28)	0.06
MCA	83 (20.5)	100 (23.0)	1.28 (0.87–1.86)	0.21	1.23 (0.75–2.02)	0.40
PCQ	56 (13.8)	60 (13.8)	1.14 (0.73–1.77)	0.58	0.76 (0.42–1.36)	0.36
Endovascular therapy	183 (45.2)	214 (49.2)	1.18 (0.90–1.54)	0.245	0.97 (0.67–1.40)	0.86

Values are expressed as mean ± 2SD or the number of patients (%). Statistically significant valuables are indicated by boldface. ICA = internal carotid artery, ACA = anterior cerebral artery, MCA = middle cerebral artery, PCQ = posterior circulation, WFNS = World Federation of Neurosurgical Societies.

## Data Availability

The raw data supporting the conclusions of this article will be made available by the authors on request.

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
