# Peer review of "Non-Saccular Aneurysm Shape as a Poor Prognostic Factor in Younger Patients with Spontaneous Subarachnoid Hemorrhage"

_jcm, 2025, doi:10.3390/jcm14124289_

Round 1

Reviewer 1 Report

Comments and Suggestions for Authors

The article deal with non-saccular aneurysms, that are more common in younger patients and in the posterior circulation. They are linked to higher rates of ischemic complications and poorer functional outcomes after subarachnoid hemorrhage compared to saccular aneurysms. This negative impact is especially pronounced in patients aged 60 or younger.

The article is well written. The statistical analysis appears to be properly conducted, and no major flaws are evident. 

Author Response

Comments 1:
The article deal with non-saccular aneurysms, that are more common in younger patients and in the posterior circulation. They are linked to higher rates of ischemic complications and poorer functional outcomes after subarachnoid hemorrhage compared to saccular aneurysms. This negative impact is especially pronounced in patients aged 60 or younger.
The article is well written. The statistical analysis appears to be properly conducted, and no major flaws are evident. 

Response 1:
Thank you for positively evaluating our manuscript. We hope our study will contribute to improving functional outcomes of subarachnoid hemorrhage caused by non-saccular aneurysms.

Reviewer 2 Report

Comments and Suggestions for Authors

Dear authors,

you compare retrospectively the prognosis of ruptured berry aneurysms with non-saccular aneurysms excluding mycotic aneurysms.  Non-saccular aneurysms / pseudo-aneurysms are a very heterogeneous entity. In order to now, about what you are speaking, it would be at least necessary to know, how many fusiform, dissecting, blister or marantic aneurysms were encountered (preferred in an additional table).

Author Response

Comment 1:

Dear authors,

you compare retrospectively the prognosis of ruptured berry aneurysms with non-saccular aneurysms excluding mycotic aneurysms.  Non-saccular aneurysms / pseudo-aneurysms are a very heterogeneous entity. In order to now, about what you are speaking, it would be at least necessary to know, how many fusiform, dissecting, blister or marantic aneurysms were encountered (preferred in an additional table).

Response 1:

As the reviewer suggested, the breakdown of non-saccular aneurysms is important to describe a very heterogeneous entity. In our original database, the non-saccular aneurysms were categorized into “dissecting”, “fusiform”, “blood blister”, and “infectious” aneurysms. “Pseudoaneurysms”, which is a histological term, were not recorded to avoid confusion. Because we excluded infectious aneurysms from the analysis, we made a table to describe the breakdown of “dissecting”, “fusiform”, and “blood blister” aneurysms and their location. (Table 2, Page 3 Line 108-109, Page 5 Line 176-179) According to the formatting error I cannot fix, I placed the Table 2 at the end of the manuscript after conclusion. Sorry for inconvenience.

Reviewer 3 Report

Comments and Suggestions for Authors

the manuscript is well written and its scientific soundness is good. the methodology is well described and the patients data is well selected. the only change that I would suggest is to be more specific in non saccular aneurysms which is too wide and as such it encompasses many entities that need different approaches. I would suggest to describe them in detail.

Author Response

Reviewer 3

Comment 1:
the manuscript is well written and its scientific soundness is good. the methodology is well described and the patients data is well selected. the only change that I would suggest is to be more specific in non saccular aneurysms which is too wide and as such it encompasses many entities that need different approaches. I would suggest to describe them in detail.

Response 1:
As the reviewer suggested, the breakdown of non-saccular aneurysms is important to describe a very heterogeneous entity. In our original database, the non-saccular aneurysms were categorized into “dissecting”, “fusiform”, and “blood blister” aneurysms. We made a table to describe the breakdown of “dissecting”, “fusiform”, and “blood blister” aneurysms and their location. (Table 2, Page 5 Line 176-179) Also, treatment modalities and ischemic complications were investigated in each category, suggesting that endovascular therapy was more often used for dissecting aneurysms and the rate of ischemic complications was comparable. We added this statement (Page 6 Line 187-190).
According to the formatting error I cannot fix, I placed the Table 2 at the end of the manuscript after conclusion. Sorry for inconvenience.

Round 2

Reviewer 2 Report

Comments and Suggestions for Authors

Dear authors

thank you for improving the article

Regarding references, no. 15 does not fit to the context, no 6,7, and 13 are self-citations without much impact, they should be should be replaced by more relevant literature or canceled.

Author Response

Comment 1:

Dear authors

thank you for improving the article

Regarding references, no. 15 does not fit to the context, no 6,7, and 13 are self-citations without much impact, they should be should be replaced by more relevant literature or canceled.

Response 1:

As the reviewer suggested, reference 15 does not fit to the context. Reference 6, 7, and 13 are self-citations without much impact. As we do not necessarily have to include these articles, we canceled them.

Reviewer 3 Report

Comments and Suggestions for Authors

I would like to thank the authors for their comments and revision. The description added to the text is suitable and accurate for acceptance.

Author Response

Reviewer 3

I would like to thank the authors for their comments and revision. The description added to the text is suitable and accurate for acceptance.

Response 1:

Thank you for your positive comments. Regarding English language, we had our manuscript checked by an English editing service “Editage” https://www.editage.jp

We believe the quality of the English is now better than the original version.